# Knowledge Model Prompting Increases LLM Performance on Planning Tasks

## Abstract

Large Language Models (LLM) can struggle with reasoning ability and procedural tasks. Many prompting techniques have been developed to assist with LLM reasoning, notably Chain-of-Thought (CoT); however, these techniques, too, have come under scrutiny as LLMs' ability to reason at all has come into question. Borrowing from the domain of cognitive and educational science, this paper investigates whether the Task-Method-Knowledge (TMK) framework can improve LLM reasoning capabilities beyond its previously demonstrated success in educational applications. The TMK framework's unique ability to capture causal, teleological, and hierarchical reasoning structures, combined with its explicit task decomposition mechanisms, makes it particularly well-suited for addressing language model reasoning deficiencies, and unlike other hierarchical frameworks such as HTN and BDI, TMK provides explicit representations of not just what to do and how to do it, but also why actions are taken. The study evaluates TMK by experimenting on the PlanBench benchmark, focusing on the Blocksworld domain to test for reasoning and planning capabilities, examining whether TMK-structured prompting can help language models better decompose complex planning problems into manageable sub-tasks. Results also highlight significant performance inversion in reasoning models. TMK prompting enables the reasoning model to achieve up to an accuracy of 97.3% on opaque, symbolic tasks (Random versions of Blocksworld in PlanBench) where it previously failed (31.5%), suggesting the potential to bridge the gap between semantic approximation and symbolic manipulation. Our findings suggest that TMK functions not merely as context, but also as a mechanism that steers reasoning models away from their default linguistic modes to engage formal, code-execution pathways in the context of the experiments.

## 1 Introduction

Large Language Models (LLMs) have been criticized by recent research for their poor reasoning and planning abilities (Valmeekam et al., 2023b; Chan, 2024; Shojaee et al., 2025). Prompt engineering has been a source of study for attempts at improving these abilities, with mixed results (Stechly et al., 2024; Bhambri et al., 2025). Evaluations of reasoning, through notable methods that allow for problem decomposition such as Chain-of-Thought (CoT) (Wei et al., 2022), have shown modest improvement in some planning domains. Other research has called into question the validity of these findings and argues that LLMs are incapable of the in-context procedural reasoning from the n-shot examples that the CoT authors claim (Stechly et al., 2024; Bhambri et al., 2025). Critics argue that LLMs are incapable of planning but instead rely on pattern matching with similar prompts for performance increases.

We investigate whether prompting using a TMK framework can improve LLM performance on planning tasks by borrowing methods from the domain of cognitive and education science (Sushri et al., 2024; Dass et al., 2025; Lum et al., 2025), while still accounting for criticisms given to previous research. In the introduction and related works sections of CoT and ReACT papers (Wei et al., 2022; Yao et al., 2022), the cognitive thought process was similarly referred to as motivation.

We introduce TMK (Task, Method, Knowledge) (Murdock & Goel, 2008), a knowledge-based self-explanation framework used successfully to explain procedural learning to students (Sushri et al., 2024), into a language model prompt to determine if TMK can improve a language model's proce-

dural planning ability. The paper seeks to extend on previous work from other fields by focusing on TMK language, which was developed as part of research into cognitive architectures in intelligent agent systems to allow systems to reason about their own processing and make any necessary adjustments (Murdock, 2001; Murdock & Goel, 2008). TMK is a representationally agnostic model authored by domain experts; JSON is one serialization used when writing the model into the LLM prompt (Dass et al., 2025). Successful use in the education domain to supplement LLM in teaching procedural learning to students, offered motivation to explore whether the TMK framework could also qualitatively improve language models' performance. Beyond raw scores, we investigate TMK's potential as an additional inference steering mechanism. A structured prompt that can steer a model's reliance on probabilistic linguistic patterns and activate its latent symbolic processing capabilities.

The hypothesis driving the experiments is assessing only raw scores. We hypothesize that a TMK-structured prompt acts as a **symbolic steering mechanism**, enabling language models to decouple logical reasoning from semantic interference. We test this by evaluating whether TMK improves planning performance specifically in the PlanBench Blocksworld domain, where semantic cues are absent (Random Blocksworld) or misleading (Mystery Blocksworld). This is to be supported by observing bias shifts from linguistic approximation to symbolic execution with TMK.

To test this hypothesis, we will use OpenAI models available at time of writing against the Plan-Bench benchmark (Valmeekam et al., 2023b). The benchmark evaluates language models on a set of planning problems, which maintains a publicly available leader board (Valmeekam, 2023) of Blocksworld problems and their variations that we will use as a comparison for our hypothesis.

Our findings indicate that TMK inclusion does increase the success of the language model in planning tasks across the flagship models. We observe improvements hold for flagship models but not models that are specifically optimized for domains or expediency. Optimization techniques such as distillation, quantization, pruning and other techniques as shown in Zhu et al. (2024) often comes at a cost of performance and can cause catastrophic forgetting (McCloskey & Cohen, 1989) which may impact planning, we discuss this further in section 5.

Across models and the various Blocksworld datasets, the TMK framework generally increased performance, with significant gains across reasoning models for random Blocksworld problems and a maximum gain of 65.8% on the o1 model for Random Blocksworld (rising from 31.5% to 97.3%). We believe the performance gains are an indication that the TMK framework is likely to demonstrate similar gains in planning tasks for other domains suggested in section 6.

## 2 RELATED WORK

The paper's research is at the intersection of two distinct areas: standardized planning benchmarks for language models and TMK prompting strategies for planning. While prior work has investigated these areas in isolation, this paper explores how integrating TMK into prompts can improve planning tasks and surpass state-of-the-art (SoTA) performance identified in recent literature for flagship models in publicly available planning benchmarks (Valmeekam, 2023).

### 2.1 PROMPTING FOR PLANNING

Prompt engineering for supporting a language model's response is a well-researched area, with CoT and ReACT, being the most notable with regard to problem decomposition (Wei et al., 2022; Yao et al., 2022). A literature survey by Sahoo et al. (2024) found over 40 prompting techniques. However, due to the wide breadth of applications, few have claimed improvement in planning. Even fewer define planning formally, where every step leading to the final answer must be correct (as opposed to approximations that sound feasible but are incorrect).

**Chain-of-Thought and Its Limitations.** While CoT showed initial promise on reasoning tasks, recent systematic evaluations reveal severe limitations for classical planning. Stechly et al. (2024) tested CoT on 261 simpler Blocksworld problems, "Table-To-Stack" configurations where every block starts on the table, without pre-defined stacks, with the goal of one single stack. Stechly et al. (2024) found that zero-shot CoT on average achieved only 1.1% improvement when compared to no CoT (baselined across GPT-4, GPT-4-Turbo, and Claude-3-Opus).

**Chain-of-Symbols (CoS)'s Informality.** CoS (Hu et al., 2023) another in-literature candidate, uses an extension of CoT using symbolic representations with 5 demonstrated (five-shot) examples. Brick World, despite the name similarity, is a distinct, simpler domain from classical Blocksworld. It uses Longest Common Sequence(LCS) to evaluate intermediate steps recorded as precision (LCS divided by length of output) and recall (LCS divided by length of ground truth) percentages, and reports the final answer match as accuracy. It lacks the formal PDDL specification and validation mechanisms required for rigorous planning evaluation.

**ReACT's Brittleness in Planning.** ReACT, which interleaves reasoning traces with action execution, faces similar challenges. A critical evaluation by Bhambri et al. (2025) found that ReACT's performance on AlfWorld and WebShop domains is minimally influenced by the reasoning trace content. Instead, success stems from "unreasonably high similarity between input example tasks and queries," requiring instance-specific examples that significantly increase human cognitive burden. When examples differ even slightly from queries (e.g., different object names, goal locations, or task types within the same domain), performance collapses.

In our paper, we deal with such criticism by following the PlanBench benchmark, which has the following requirement: (1) The leader board was derived from zero-shot and one-shot examples, which discounts highly identical example similarity as a factor to planning ability. (2) A correct answer is not just the final state, but also the entirety of the given reasoning process. These two requirements address criticisms given in this section. For this paper, we will only focus on Blocksworld and OpenAI models; however, we believe that our results indicate a strong direction for future research.

## 2.2 PLANBENCH

Existing literature shows that language models struggle with classical planning tasks (Valmeekam et al., 2023b; Chan, 2024; Shojaee et al., 2025). Planning requires every action to be formally valid ("close enough" approximations fail verification) and is fundamentally distinct from the question-answering and commonsense reasoning tasks in ReACT and CoT (Wei et al., 2022; Yao et al., 2022), where next-token prediction patterns can often derive correct final answers through approximating intermediate as described in section 2.1. This paper utilizes PlanBench (Valmeekam et al., 2023a), an extensible benchmark based on International Planning Competition (IPC) domains that verifies language model outputs using classical planning tools (Helmert, 2006; Howey et al., 2004) without providing the models access to those tools at inference time. Rather, it uses those tools to robustly validate plans generated by language models. Unlike commonsense and arithmetic reasoning benchmarks, PlanBench tests the planning ability of a language model and requires formal symbolic correctness validated by automated planners and plan validators.

Crucially, PlanBench introduces domain obfuscation to decouple a language model's reasoning ability from its pre-trained linguistic priors. As shown in Table 1, the benchmark includes three variants:

- **Classic:** Uses canonical English labels (e.g., pick up, stack), allowing models to rely on semantic memory.

- **Mystery:** Maps actions to semantically distinct but unrelated words (e.g., attack, feast). This tests if the model can reason using the provided rules rather than semantic associations.

- **Random:** Replaces labels with opaque alphanumeric strings. This steers the model to rely more on symbolic manipulation.

For the scope of this paper, we focus on Blocksworld across these three to give some initial insight into how TMK impacts planning in language models. This selection provides a baseline for language models without TMK, allowing us to isolate whether the TMK framework improves planning tasks in Blocksworld and its more challenging variations.

## 2.3 THE TMK FRAMEWORK

The TMK (Task, Method, Knowledge) framework is a formal knowledge representation language originally developed to model the teleology of intelligent agents (Murdock, 2001). It hierarchically decomposes a system into three components: (1) **Tasks** (goals and conditions), (2) **Methods** (procedural mechanisms), and (3) **Knowledge** (domain concepts and relationships). This explicit

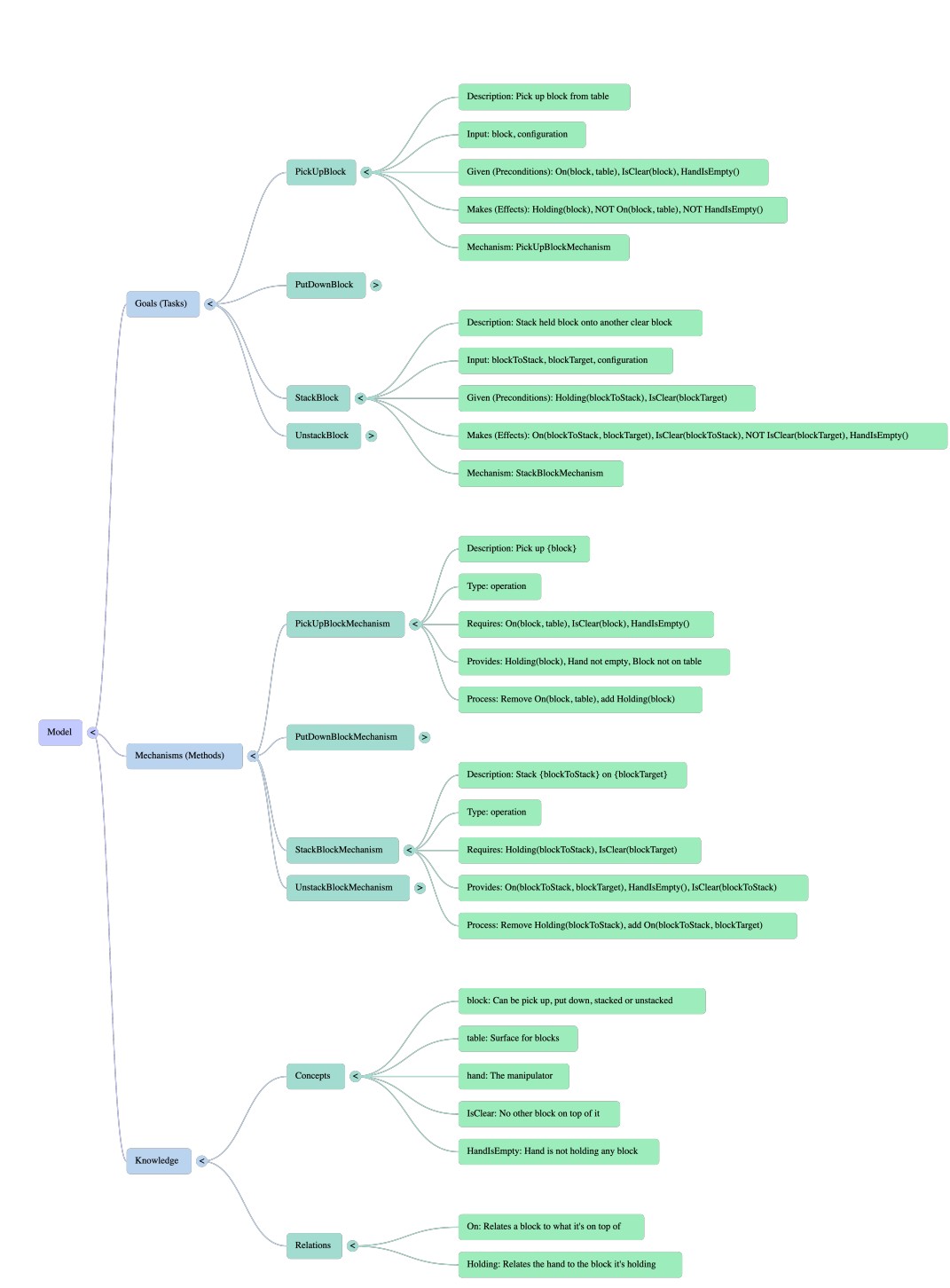

Figure 1: Expansion of an example Task, Method and Knowledge structure, where goals are teleologically connected to methods by which they are achieved using their knowledge of the environment or domain (Blocksworld in this example). See OSF link for detailed expansion: (Anonymous, 2025)

decomposition enables the capture of not only *what* to do and *how* to do it, but crucially, *why* actions are taken (teleology), which we suspect can help ground a model's reasoning and planning.

While other hierarchical task decomposition frameworks exist, such as Belief-Desire-Intention (BDI) (Georgeff et al., 1998) and Hierarchical Task Network (HTN) (Erol et al., 1994), TMK is distinct in its emphasis on teleological and causal self-explanation. Like TMK, these frameworks are candidates for symbolic interpreters and may offer future potential for dual use within both prompts and systems with symbolic solvers. However, TMK's specific architectural focus on the "why" of actions makes it uniquely applicable to our study.

The framework provides a structured basis for reasoning by breaking available actions into simpler, manageable components, a process that mirrors cognitive decomposition (Correa et al., 2023). Consequently, TMK's unique ability to capture causal and teleological structures makes it well-suited for addressing the planning deficiencies often observed in language models. It achieves this by explicitly defining clear steps (methods), the relationships between them (knowledge), and the goals driving them (tasks).

Prior research has demonstrated that language model outputs often favor declarative knowledge over procedural steps (Dass et al., 2025). This preference contributes to the poor planning performance described in section 2.2, evidence in Valmeekam (2023). Because TMKs are authored by domain experts specifically to facilitate procedural learning through causal reasoning, we hypothesize that prompting with TMK can equip a language model with the procedural and causally grounded context preferred by experts (Lum et al., 2025). Validating this hypothesis is the primary investigation of this paper. Furthermore, task decomposition was essential for reasoning in pre-LLM systems (Murdock, 2001; Georgeff et al., 1998; Erol et al., 1994). Given the documented struggles of modern LLMs with autonomous reasoning (Valmeekam et al., 2023a; Chan, 2024; Shojaee et al., 2025), applying the TMK framework to prompt engineering offers a promising avenue for enhancing LLM planning capabilities.

## 3 METHODS

### 3.1 DESIGNING A MODEL WITH THE TMK FRAMEWORK

The TMK knowledge representation framework has three components: (1) Task describes the goal of a system, (2) Method describes mechanisms by which the goal is achieved, and (3) Knowledge defines the domain ontology required to interpret tasks, mechanisms and support them (Murdock, 2001; Murdock & Goel, 2008; Goel & Rugaber, 2017). For the purpose of this paper, the authors converted the domain knowledge from plain text to TMK by breaking down each possible step within Blocksworld into methods. The authors kept to a three-layer (as shown in Fig. 1) hierarchy decomposition to maintain the conciseness of the TMK structure to keep within effective context windows (Liu et al., 2024); however, there is no limit to the number of layers for more complex procedural goals to be decomposed or the level of intricacy to be included as problem complexity and context windows grows. For instance, there can be a plan verification (i.e. verifyPlan and CheckGoalAchieved goals) that themselves can be further broken down much deeper down to actually verifying the sequence of correct steps using known algorithms, similar to verifiers and solvers such as VAL (Howey et al., 2004) and fast downward (Helmert, 2006) respectively. TMK enables calibration of the level of abstraction depending on the use case.

#### 3.1.1 TASK, GIVEN BY ITS GOALS (THE WHY)

Within each goal, fields such as *name* and *description* provide human-oriented identifiers and summaries, while input parameters and output parameters make data flow explicit. There are also the *given* and *makes* clauses to encode pre- and post-conditions of the goals, such as "holding" or "is clear". Finally, there is also a *Mechanism* field that links the goal to a method, connecting "why" (goal) and "how" (mechanism). These fields in the TMK framework help guide designers and provide a structure as they build out their TMK.

### 3.1.2 METHOD, GIVEN BY ITS MECHANISMS (THE HOW)

Mechanisms correspond to goals, and they include similar *description*, *input*, and *output* fields but differ in having *require* and *provides* clauses to differentiate logical requirements of concepts and relations The *require* and *provides* clauses alongside an operation within the process field document the logical causal updates, allowing the mechanism's post-condition to satisfy a parent goal, which in our case is picking up a block.

### 3.1.3 KNOWLEDGE, DEFINES THE DOMAIN-SPECIFIC CONCEPTS AND RELATIONSHIPS, GOALS, AND MECHANISMS USED

Knowledge defines the ontology that parameter names and logical conditions reference in the TMK, ensuring consistent interpretation across goals and mechanisms. This combination is used in the experiment to define the domain knowledge of a procedural knowledge and reasoning question, giving the AI models a description of the decomposed task as input.

### 3.1.4 TMK DESIGN, PLACEMENT AND IMPLEMENTATION

Our TMK prompt replaces the domain portion of the PlanBench prompt shown in Valmeekam et al. (2023b) with a TMK formatted in JSON, using the one-shot example and querying it for consistency. This can also be compared in the prompt examples in appendix A.

It is important to note that the process of designing and building a TMK is iterative and can be decomposed deeper or abstracted further, as evident in Murdock (2001). In this paper, it is kept to three layers for ease of discussion and experimentation. The level of abstraction is left to the designer, who can also utilize the framework to structure their thinking.

There are also differences in TMK in classic blocksworld, mystery blocksworld, and random blocksworld (see in OSF link, Anonymous (2025)). All prompts and system prompts are also included in the results file. Efforts are made to keep it as similar as possible, given constraints required by the different types of Blocksworld datasets. This difference is also consistent in the PlanBench prompts, where 'random' and 'mystery' Blocksworld were using true and false descriptions, but not the classical Blocksworld prompt, as Valmeekam et al. (2023a)'s prompt evolved.

### 3.2 EVALUATION ON PLANBENCH

As part of this paper, we evaluated the TMK against the public PlanBench benchmarks found on Valmeekam (2023). There are a few key differences between our method and the one described as part of the benchmark. One of which is that our testing is one-shot while the Valmeekam (2023) is zero-shot. This difference is inconsequential to our findings for the following reasons:

1. **Baseline Rigor and Formatting:** The first is that the original PlanBench paper (Valmeekam et al., 2023a) consist entirely of one-shot questions while the publicly available leader board, Valmeekam (2023) is of zero-shot. Given this mix, we made the decision to compare against the higher value (zero-shot) for rigor. For TMK, one-shot prompts also allowed us to conform TMK outputs exactly to what Valmeekam (2023) expects while reducing formatting instructions, minimizing non-TMK differences in prompts.

2. **Precedence and Performance:** The papers listed on Valmeekam (2023) reported only the zero-shot case, we suspect it is because as observed in the data, zero-shot did better than the one-shot for those listed papers (for plain text) (Valmeekam, 2023; Valmeekam et al., 2023c). Literature (Kojima et al., 2022) also confirms that LLMs are zero shot reasoners and this is additionally backed by our sample testing of one-shot examples in plain text (see results and code in Anonymous (2025), OSF link).

3. **Nature of the Example:** Lastly, although we offer a one-shot example, that example is random and not tailored to the problem at hand as is often the case with similar research.

The Valmeekam (2023) code at the time of writing required update in the extracting random blocksworld to be comparable with the ground truth. Our experiment thus also added new code (included in Anonymous (2025)) to the extraction criteria which was applied for random blocksworld

data set. It focuses on the sequence of execution and order of objects after the actions, which are the key set of criteria to determine the ability of the LLM to plan.

The authors also made the choice to not get perturbed by the stochastic language model responses that can sometimes include: (1) symbols (like "-", "_" ect.), (2) words (like "o", "obj") instead of "object"(blocks) and (3) plan steps like "stack object b from object a", (translated to "Overcome object b from object a" and "2ijg9q8swj2shjel stack object b from object a" for mystery and random domains respectively). Our enhanced extraction function prevents these from getting evaluated as incorrect because even if words like "object", "from" and symbols like "-", "_", show up in the response of a model, their sequence and ground truth action along with their order match.

This is rare in classic blocksworld, but seems to be an artifact evident within random blocksworld domains since language models often have a bias towards generating grammatically correct, human understandable, english sentences (Hu et al., 2024). The authors believe these stochastic errors do not take away from the ability to assess if language models can plan and reason, which is consistent within ICAPS (International Conference on Automated Planning and Scheduling) language reference documents (Fern, 2008) and importantly, similar to extraction examples listed in papers that investigated PlanBench further (Valmeekam et al., 2023c).

We further confirm these findings by running the PlanBench benchmark for newer models for which it has not been reported. In these cases, similar to the paper by the other authors, the one-shot case was worse than the zero-shot case (shown in results file in OSF, Anonymous (2025)).

## 4 EXPERIMENT

The results of the experiment is consolidated in Table 2. Interestingly, while TMK improvements are evident there is also evidence that under the methods of this experiment, more robust benchmarks for planning are needed for LRMs (large reasoning models) such as o1 and GPT-5, as they perform particularly well on the blocksworld domain with TMK prompt significantly outperforming SoTA in the random blocksworld domain.

### 4.1 TMK OUTPERFORMED PLAIN TEXT

At a high level, the experiments show that all flagship models with domain prompts replaced by TMK structures perform better. The authors posit the that optimization processes (i.e. quantization, pruning, distillation) that resulted in o1-mini's vastly reduced inference time, cost, and training with a mathematical skew (OpenAI, 2025) may have caused it to be an outlier (further discussed in section 5). Though without access to information beyond what was released by OpenAI, it remains difficult to ascertain why. The majority of the models have also shown a marked improvement (in bold) in different blocksworld sub-domains when prompted using the TMK framework with gains up to 65.8% in o1 on the random dataset.

### 4.2 COMPARING BETWEEN MYSTERY AND RANDOM BLOCKSWORLD

The experiments reveal insight on LRMs that justifies further investigation. Notably, we observe a strong *performance inversion* in the reasoning models.

In the plain text baseline, o1 follows the typical pattern of language models, performing significantly better on Mystery (74.3%) than on Random (31.5%). This suggests that without structure, the model relies on semantic cues (even misleading ones) to guide its reasoning. However, under TMK prompting, this relationship flips: o1 scores on Random (97.33%), surpassing its Mystery (83.3%).

This suggests that the TMK framework acts as a symbolic scaffold. By imposing a rigid, code-like structure, TMK appears to shift the model's inference strategy away from linguistic approximation and toward formal symbolic manipulation, a mode that operates most effectively when the tokens (e.g., "1jpkithdyjmlikck", aka "pick up") are devoid of pre-trained semantic associations.

Conversely, the smaller o1-mini model does not show this inversion benefit in the Mystery domain (dropping to 16.83%). We hypothesize that unlike flagship reasoning models, o1-mini lacks the capacity to resolve the conflict between the rigid TMK structure and the semantic interference of the Mystery vocabulary, resulting in the degradation observed in Table 2.

Table 1: Blocksworld predicate and action name correspondences across PlanBench domain variants: Classic (canonical names), Mystery (semantically meaningful obfuscations), and Random (opaque identifiers) used to evaluate planning under reduced linguistic cues while preserving a one to one mapping of planning semantics (Valmeekam et al., 2023a).

| Classic | Mystery | Random |
|---|---|---|
| **Predicates** | | |
| empty hand | Harmony | 3covmuy4yrjthijd |
| holding | Pain | gk5asm3f7u1fekpj |
| on table | Planet | 51nbwlachmfartjn |
| on | Object Craves | 4dmf1cmtyxgsp94g |
| clear | Province | aqcjuuehivl8auwt |
| **Actions** | | |
| pick up | Attack | 1jpkithdyjmlikck |
| put down | Succumb | 9big8ruzarkkquyu |
| stack | Overcome | 2ijg9q8swj2shjel |
| unstack | Feast | xptxjrdkbi3pqsqr |

Table 2: Accuracy (percentage of fully correct plans with complete, stepwise reasoning) on PlanBench Blocksworld variants, Classic, Mystery, and Random comparing plain-text prompts (best of sampled Zero & One shot) and TMK structured prompts (One shot); tasks and evaluation follow PlanBench definitions (Valmeekam et al., 2023a). Bold values indicate significantly improvements (*Note: o1Preview has been depreciated and replaced by o1. Results extracted from Valmeekam (2023))

| Model | Type | Plain Text (%) | TMK (%) |
|---|---|---|---|
| **LLM (Large Language Models)** | | | |
| GPT4 | Classic | 34.6 | 39.7 |
| | Mystery | 0 | 3.8 |
| | Random | 0 | 4.17 |
| GPT4o | Classic | 35.5 | **45.3** |
| | Mystery | 0 | **5.5** |
| | Random | 0.83 | 4.83 |
| **LRM (Large Reasoning Models)** | | | |
| o1mini | Classic | 56.7 | 57 |
| | Mystery | 19.1 | *16.83* |
| | Random | 9.33 | **27.0** |
| o1preview* | Classic | 97.8 | NA |
| | Mystery | 52.8 | NA |
| | Random | 37.3 | NA |
| o1 | Classic | 95.7 | 98.5 |
| | Mystery | 74.3 | **83.3** |
| | Random | 31.5 | **97.33** |
| GPT5 | Classic | 99.3 | 99.7 |
| | Mystery | 98.1 | 98.3 |
| | Random | 92.5 | **99.0** |

## 5 DISCUSSION

The experimental results in Table 2 demonstrate that integration of a TMK framework into an LLM prompt improves procedural capabilities in a language model. The improvements are not merely incremental in some cases as in o1, the performance on random blocksworld dataset can transform a competent (31.5%) into a SoTA result (97.3%) and for o1-mini, near failure (9.33%) into a competent result (27%). This discussion explores the underlying reasons. The paper argues that the TMK structure acts as a "cognitive scaffold" that decomposes tasks on behalf of language models. The authors posit that the TMK structure can help shift a model's inference processes towards a more formal symbolic mode of operations, more akin to its code generation training data set than plaintext data set. The existence of a code or textual skew in latent space has only recently been reported by Chen et al. (2024), also referencing PlanBench.

### 5.1 CAN LLMs REASON BETTER GIVEN PROMPTING?

Some of the criticisms (Stechly et al., 2024; Bhambri et al., 2025) of language model reasoning include: (1) Papers claiming that prompting increasing reasoning ability in LLMs often use N-shot examples from which the LLM engages in pattern matching, (2) In cases where LLM show CoT as part of their reasoning, the CoT often contradicts the final answer given, and (3) Even in Zero-Shot prompting cases, LLMs do not show significant ability to plan across different domains or models.

Our research shows gains in LLM planning ability while, for the most part, avoiding the criticism above, through taking the following precautions where applicable, respectively: (1) We tested and demonstrated learning gains in only the one-shot case. In the one-shot case, we provided one constituent example, but unlike many of the papers criticized, our example does not match the problem in problem length or block description. Additionally, we demonstrate that in the PlanBench dataset, the zero-shot case often outperforms the one-shot case for plan text, reinforcing that it is the TMK, not the given example, that is creating the gains in accuracy. (2) We evaluate the entire explana-

tion as part of our results, meaning every step of the planning task must be correct in order to be considered. (3) We show for the blocksworld domain, TMK prompting increases the correctness of LLM reasoning across models and subdomains, although only incrementally. Further research is needed to know if TMK prompting is useful across domains. This shows that in the TMK language of describing tasks, methods, and knowledge, along with their functions, descriptions, input, output, pre- and post-conditions that can be used by LLMs to improve their course of action.

## 5.2 EXPLANATIONS FOR ACCURACY GAINS

The authors of this paper have a hypothesis in two parts about why TMK assists in LLM planning, the first relating to code in the training data, and the second as tied to cognitive and educational science

### 5.2.1 STEERING BETWEEN CODE EXECUTION AND TEXTUAL REASONING

Language models are often trained with both code and plain text data (Langlais et al., 2025; Aryabumi et al., 2024). With the TMK framework with nested parenthesis, keywords, explicit variable assignment is structurally analogous to more formal syntax of programming languages. This structural similarity likely prompts the model to shift its inference strategy from a linguistic mode to a more code-like symbolic mode of internal token manipulation.

It is feasible that the superior performance of TMK structure prompts can be attributed to their ability to activate formal reasoning pathways inherent in a model's code training data. Unlike plain text that a language model was trained on, code training data will naturally contain variables that, while may not look exactly like "1jpkithdyjmlikck" (pick up), can allow a language model to consider and use "1jpkithdyjmlikck" as an arbitrary identifier. This identifier can substitute the word "pick up" and be used in various logical or non-linguistically sequential context that allows attention tokens within transformer architectures (Vaswani et al., 2017) to look further than if it were linguistic (English) sentences. In a more code analogous structure, attention must track long-range dependencies such as linking a variable's definition to its uses many lines later. The hypothesis is reinforced by the generally greater improvements in LRM scores compared to LLMs from TMK structured prompts. It shows that LRM with longer test time inference can better capitalize on the TMK hierarchical and teleological structures. Compared to plain text, it is reasonable to deduce that the inference (reasoning) tokens of TMK should contain more code-like structures, given that TMK is in JSON format. This should be tested in models that have transparent reasoning tokens as part of future work. Overall, a TMK-formatted prompt appears to be effective at unlocking latent capabilities of LRM for procedural tasks.

This aligns with recent findings by Chen et al. (2024), who demonstrate that textual reasoning has inherent limitations for logic and optimization tasks, often necessitating a shift to code-based execution for accuracy. The TMK framework effectively acts as a steering mechanism, forcing the model to bypass its default textual reasoning pathways, which are susceptible to the semantic interference observed in the Mystery domain—and engage the code-execution reasoning pathways that Chen et al. (2024) identify as superior for symbolic tasks.

The 'Performance Inversion' observed in the o1 model—where symbolic tasks (Random) become significantly easier than semantic ones (Mystery) under TMK, serves as empirical validation of this steering effect. If TMK were simply providing additional context, we would expect uniform gains across domains. Instead, the reversal of domain difficulty indicates a fundamental shift in the underlying reasoning modality: the prompt successfully steers the model out of the semantic interference zone and into a robust symbolic manipulation zone."

### 5.2.2 COGNITIVE SCAFFOLDING

Within the literature review of cognitive and educational science, TMK encourages LLMs to return procedural explanations when compared to unstructured output (Dass et al., 2025). These more procedurally focused explanations are generally preferred by domain experts, but novice learners tend to prefer the more knowledge-focused conversational style of unstructured text (Lum et al., 2025). Under Bloom's taxonomy of learning, procedural understanding is considered a higher level of learning compared to factual knowledge-focused answers; however, this type of learning is considered more

cognitively demanding under cognitive load theory (Krathwohl, 2002). Given this, the preference for unstructured knowledge-focused responses in LLMs may be a side effect of the worked example effect (Kirschner et al., 2010). The worked example effect is a well-known effect in education science where, for novice learners, having worked examples, that is, examples where much of the problem is solved, minus key steps that are being taught, has been shown to be immensely effective for novice learners. Despite this, novices have been shown to dislike using worked examples in their learning. For this reason, TMK prompting may encourage LLMs to produce more procedural reasoning explanations, which are more likely to be made by experts, increasing their planning abilities. From the perspective of LLM reasoning critics, while TMK prompting does not offer n-shot solutions to which the LLM then pattern matches, it instead offers expert knowledge to which the LLM pattern matches. That is, prompting with information structured in ways preferred by experts in procedural tasks allows the LLM to perform better on those tasks.

### 5.3 LIMITATIONS

Our research only investigates improving language model reasoning and planning within the domain of Blocksworld and its variant found in the PlanBench benchmark. Future work should expand into Logistics problems (within PlanBench) or other extensible domains and other families of models. Another key component of planning that introduces distinct challenges is multi-agent coordination and path finding. Investigating whether TMK's efficacy transfers to these distinct planning patterns would help validate the generalizability of our findings beyond stacking tasks.

## 6 CONCLUSION

Our work indicates a general improvement of plan correctness on PlanBench Blocksworld variants. Improvements across (LLM and LRM) models tested were shown for Classic Blocksworld and Random Blocksworld, with Mystery Blocksworld showing improvement on all but o1-mini model. We also saw our greatest increase in model performance in the flagship o1 model in Random Blocksworld with a 65.8% increase in planning accuracy. Crucially, this result represents a fundamental 'performance inversion': whereas standard prompting struggles with opaque symbolic tasks (Random) compared to semantic ones (Mystery), TMK prompting reversed this dynamic for the o1 model, enabling it to outperform its own semantic baseline.

This confirms that TMK acts as a symbolic scaffold, effectively steering reasoning models toward formal code-like manipulation when semantic cues are absent. Improvements were consistent across flagship models, however, the smaller o1-mini model proved to be an outlier, showing regression in the Mystery domain, likely due to capacity limitations in resolving semantic interference. Larger increases were demonstrated for models that performed worse on Blocksworld in general, but even models such as GPT-5 which demonstrated a high starting success rate, still saw non-trivial increases. For the LRMs, a more robust dataset or a different domain may be needed, as while TMK did show increases in accuracy on planning tasks, their high starting success rate leaves little room for improvement. We theorize this increase is due to the TMK prompting structure that intrinsically allows language models to defer to a more code-adjacent inference. This contrasts with one that is prompted with plain text drawing on a more linguistically adjacent inference.

There was a singular notable case where TMK prompting caused a decrease in accuracy for o1-mini that warrants further investigation, although the authors theorize it is caused by semantic overload. Determining what about the TMK framework causes an increase in random and classic blocksworld but a decrease in mystery would be an interesting path for future research. Additionally, across subdomains, we also noted that the introduction of TMK shifted a pattern in plain text where LLMs often did better on mystery blocksworld when compared to random blocksworld. This could also be a side effect of TMK, causing semantic overload.

For future work, we hope to investigate other domains, as well as evaluate how well TMK performs when compared to other knowledge models such as BDI and HTNs. Our research gives some indication that prompting can increase the accuracy of LLMs in reasoning tasks within the PlanBench blocks world variant, while avoiding the common criticism of previous research; however, the cause of that increase is left to future work.

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

# A APPENDICES

**Contents**

## A.1 ONE SHOT CLASSIC BLOCKSWORLD WITH PLAIN TEXT EXAMPLE

**One shot Classic Blocksworld with Plain Text**

```
I am playing with a set of blocks where I need to arrange the blocks into stacks. Here
↪  are the actions I can do

Pick up a block
Unstack a block from on top of another block
Put down a block
Stack a block on top of another block

I have the following restrictions on my actions:
I can only pick up or unstack one block at a time.
I can only pick up or unstack a block if my hand is empty.
I can only pick up a block if the block is on the table and the block is clear. A block
↪  is clear if the block has no other blocks on top of it and if the block is not picked
↪  up.
I can only unstack a block from on top of another block if the block I am unstacking was
↪  really on top of the other block.
I can only unstack a block from on top of another block if the block I am unstacking is
↪  clear.
Once I pick up or unstack a block, I am holding the block.
I can only put down a block that I am holding.
I can only stack a block on top of another block if I am holding the block being stacked.
I can only stack a block on top of another block if the block onto which I am stacking
↪  the block is clear.
Once I put down or stack a block, my hand becomes empty.

[STATEMENT]
As initial conditions I have that, the red block is clear, the blue block is clear, the
↪  yellow block is clear, the hand is empty, the blue block is on top of the orange
↪  block, the red block is on the table, the orange block is on the table and the yellow
↪  block is on the table.
My goal is to have that the orange block is on top of the blue block.

My plan is as follows:

[PLAN]
unstack the blue block from on top of the orange block
put down the blue block
pick up the orange block
stack the orange block on top of the blue block
[PLAN END]

[STATEMENT]
As initial conditions I have that, the red block is clear, the yellow block is clear, the
↪  hand is empty, the red block is on top of the blue block, the yellow block is on top
↪  of the orange block, the blue block is on the table and the orange block is on the
↪  table.
```

```
My goal is to have that the orange block is on top of the red block.

My plan is as follows:

[PLAN]
```

## A.2 Zero shot Classic Blocksworld with Plain Text Example

**Zero shot Classic Blocksworld with Plain Text**

```
I am playing with a set of blocks where I need to arrange the blocks into stacks. Here
↪  are the actions I can do

Pick up a block
Unstack a block from on top of another block
Put down a block
Stack a block on top of another block

I have the following restrictions on my actions:
I can only pick up or unstack one block at a time.
I can only pick up or unstack a block if my hand is empty.
I can only pick up a block if the block is on the table and the block is clear. A block
↪  is clear if the block has no other blocks on top of it and if the block is not picked
↪  up.
I can only unstack a block from on top of another block if the block I am unstacking was
↪  really on top of the other block.
I can only unstack a block from on top of another block if the block I am unstacking is
↪  clear.
Once I pick up or unstack a block, I am holding the block.
I can only put down a block that I am holding.
I can only stack a block on top of another block if I am holding the block being stacked.
I can only stack a block on top of another block if the block onto which I am stacking
↪  the block is clear.
Once I put down or stack a block, my hand becomes empty.

[STATEMENT]
As initial conditions I have that, the red block is clear, the blue block is clear, the
↪  yellow block is clear, the hand is empty, the blue block is on top of the orange
↪  block, the red block is on the table, the orange block is on the table and the yellow
↪  block is on the table.
My goal is to have that the orange block is on top of the blue block.

What is the plan to achieve my goal? Just give the actions in the plan.
```

## A.3 One shot Classic Blocksworld with TMK Prompt Example

**One shot Classic Blocksworld with TMK**

```
You must adhere strictly to the JSON below, paying attention to the rules, ensuring to
↪  use only legal moves to achieve the final plan.
{
   "Goals":[
      {
         "name":"PickUpBlock",
         "description":"Pick up a block from the table.",
         "inputParameters":[
            "block",
            "configuration"
         ],
         "outputParameters":[
            "newConfiguration"
         ],
         "given":[
            "On(block, table)",
            "IsClear(block)",
            "HandIsEmpty()"
         ],
         "makes":[
            "Holding(block)",
            "NOT On(block, table)",
            "NOT HandIsEmpty()"
         ],
         "mechanism":"PickUpBlockMechanism"
      },
```

```
        {
            "name":"PutDownBlock",
            "description":"Put down a held block onto the table.",
            "inputParameters":[
                "block",
                "configuration"
            ],
            "outputParameters":[
                "newConfiguration"
            ],
            "given":[
                "Holding(block)"
            ],
            "makes":[
                "On(block, table)",
                "IsClear(block)",
                "HandIsEmpty()"
            ],
            "mechanism":"PutDownBlockMechanism"
        },
        {
            "name":"StackBlock",
            "description":"Stack a held block onto another clear block.",
            "inputParameters":[
                "blockToStack",
                "blockTarget",
                "configuration"
            ],
            "outputParameters":[
                "newConfiguration"
            ],
            "given":[
                "Holding(blockToStack)",
                "IsClear(blockTarget)"
            ],
            "makes":[
                "On(blockToStack, blockTarget)",
                "IsClear(blockToStack)",
                "NOT IsClear(blockTarget)",
                "HandIsEmpty()"
            ],
            "mechanism":"StackBlockMechanism"
        },
        {
            "name":"UnstackBlock",
            "description":"Unstack a block from on top of another block.",
            "inputParameters":[
                "blockToUnstack",
                "blockFrom",
                "configuration"
            ],
            "outputParameters":[
                "newConfiguration"
            ],
            "given":[
                "On(blockToUnstack, blockFrom)",
                "IsClear(blockToUnstack)",
                "HandIsEmpty()"
            ],
            "makes":[
                "Holding(blockToUnstack)",
                "IsClear(blockFrom)",
                "NOT On(blockToUnstack, blockFrom)"
            ],
            "mechanism":"UnstackBlockMechanism"
        }
    ],
    "Mechanisms":[
        {
            "name":"PickUpBlockMechanism",
            "description":"Pick up {block}.",
            "inputParameters":[
                "block",
                "configuration"
            ],
            "outputParameters":[
                "newConfiguration"
            ],
            "type":"operation",
            "requires":[
                "On(block, table)",
                "IsClear(block)",
```

```
          "HandIsEmpty()"
     ],
     "provides":[
          "Holding(block)",
          "Hand not empty",
          "Block not on table"
     ],
     "process":"Remove On(block, table), add Holding(block), set hand state"
},
{
     "name":"PutDownBlockMechanism",
     "description":"Put down {block}.",
     "inputParameters":[
          "block",
          "configuration"
     ],
     "outputParameters":[
          "newConfiguration"
     ],
     "type":"operation",
     "requires":[
          "Holding(block)"
     ],
     "provides":[
          "On(block, table)",
          "HandIsEmpty()",
          "IsClear(block)"
     ],
     "process":"Remove Holding(block), add On(block, table), clear hand state"
},
{
     "name":"StackBlockMechanism",
     "description":"Stack {blockToStack} on {blockTarget}.",
     "inputParameters":[
          "blockToStack",
          "blockTarget",
          "configuration"
     ],
     "outputParameters":[
          "newConfiguration"
     ],
     "type":"operation",
     "requires":[
          "Holding(blockToStack)",
          "IsClear(blockTarget)"
     ],
     "provides":[
          "On(blockToStack, blockTarget)",
          "HandIsEmpty()",
          "IsClear(blockToStack)"
     ],
     "process":"Remove Holding(blockToStack), add On(blockToStack, blockTarget),
     ↪    update clear states"
},
{
     "name":"UnstackBlockMechanism",
     "description":"Unstack {blockToUnstack} from {blockFrom}.",
     "inputParameters":[
          "blockToUnstack",
          "blockFrom",
          "configuration"
     ],
     "outputParameters":[
          "newConfiguration"
     ],
     "type":"operation",
     "requires":[
          "On(blockToUnstack, blockFrom)",
          "IsClear(blockToUnstack)",
          "HandIsEmpty()"
     ],
     "provides":[
          "Holding(blockToUnstack)",
          "IsClear(blockFrom)"
     ],
     "process":"Remove On(blockToUnstack, blockFrom), add Holding(blockToUnstack),
     ↪    update states"
}
],
"Knowledge":{
     "Concepts":[
```

```
            {
                "name":"block",
                "description":"A block in the blocks world that can be pick up, put down,
                ↪   stacked or unstacked"
            },
            {
                "name":"table",
                "description":"The surface where blocks can be pick up, put down or unstacked
                ↪   onto"
            },
            {
                "name":"hand",
                "description":"The manipulator that can pick up, put down, stacked or
                ↪   unstacked blocks"
            },
            {
                "name":"IsClear",
                "description":"A block is clear if no other block is on top of it"
            },
            {
                "name":"HandIsEmpty",
                "description":"The hand is not holding any block"
            }

        ],
        "Relations":[
            {
                "name":"On",
                "description":"Relates a block to what it's on top of (another block or
                ↪   table)"
            },
            {
                "name":"Holding",
                "description":"Relates the hand to the block it's holding"
            }
        ]
    }
}

Below, within [Plan] and [Plan End], is the format you will use for the answer. The first
↪   one is an example. Focus on only the second plan.

[STATEMENT]
As initial conditions I have that, the red block is clear, the blue block is clear, the
↪   yellow block is clear, the hand is empty, the blue block is on top of the orange
↪   block, the red block is on the table, the orange block is on the table and the yellow
↪   block is on the table.
My goal is to have that the orange block is on top of the blue block.

My plan is as follows:

[PLAN]
unstack the blue block from on top of the orange block
put down the blue block
pick up the orange block
stack the orange block on top of the blue block
[PLAN END]

[STATEMENT]
As initial conditions I have that, the red block is clear, the yellow block is clear, the
↪   hand is empty, the red block is on top of the blue block, the yellow block is on top
↪   of the orange block, the blue block is on the table and the orange block is on the
↪   table.
My goal is to have that the orange block is on top of the red block.

My plan is as follows:

[PLAN]
```

## A.4   ONE SHOT MYSTERY BLOCKSWORLD WITH PLAIN TEXT EXAMPLE

**One shot Mystery Blocksworld with Plain Text**

```
I am playing with a set of objects. Here are the actions I can do

Attack object
Feast object from another object
Succumb object
```

```
Overcome object from another object

I have the following restrictions on my actions:
To perform Attack action, the following facts need to be true: Province object, Planet
↪   object, Harmony.
Once Attack action is performed the following facts will be true: Pain object.
Once Attack action is performed the following facts will be false: Province object,
↪   Planet object, Harmony.
To perform Succumb action, the following facts need to be true: Pain object.
Once Succumb action is performed the following facts will be true: Province object,
↪   Planet object, Harmony.
Once Succumb action is performed the following facts will be false: Pain object.
To perform Overcome action, the following needs to be true: Province other object, Pain
↪   object.
Once Overcome action is performed the following will be true: Harmony, Province object,
↪   Object Craves other object.
Once Overcome action is performed the following will be false: Province other object,
↪   Pain object.
To perform Feast action, the following needs to be true: Object Craves other object,
↪   Province object, Harmony.
Once Feast action is performed the following will be true: Pain object, Province other
↪   object.
Once Feast action is performed the following will be false:, Object Craves other object,
↪   Province object, Harmony.

[STATEMENT]
As initial conditions I have that, object b craves object c, harmony, planet object a,
↪   planet object c, planet object d, province object a, province object b and province
↪   object d.
My goal is to have that object c craves object b.

My plan is as follows:

[PLAN]
feast object b from object c
succumb object b
attack object c
overcome object c from object b
[PLAN END]

[STATEMENT]
As initial conditions I have that: object a craves object b, object d craves object c,
↪   harmony, planet object b, planet object c, province object a and province object d.
My goal is for the following to be true: object c craves object a.

My plan is as follows:

[PLAN]
```

## A.5 ZERO SHOT MYSTERY BLOCKSWORLD WITH PLAIN TEXT EXAMPLE

**Zero shot Mystery Blocksworld with Plain Text**

```
I am playing with a set of objects. Here are the actions I can do

Attack object
Feast object from another object
Succumb object
Overcome object from another object

I have the following restrictions on my actions:
To perform Attack action, the following facts need to be true: Province object, Planet
↪   object, Harmony.
Once Attack action is performed the following facts will be true: Pain object.
Once Attack action is performed the following facts will be false: Province object,
↪   Planet object, Harmony.
To perform Succumb action, the following facts need to be true: Pain object.
Once Succumb action is performed the following facts will be true: Province object,
↪   Planet object, Harmony.
Once Succumb action is performed the following facts will be false: Pain object.
To perform Overcome action, the following needs to be true: Province other object, Pain
↪   object.
Once Overcome action is performed the following will be true: Harmony, Province object,
↪   Object Craves other object.
Once Overcome action is performed the following will be false: Province other object,
↪   Pain object.
```

```
To perform Feast action, the following needs to be true: Object Craves other object,
↪  Province object, Harmony.
Once Feast action is performed the following will be true: Pain object, Province other
↪  object.
Once Feast action is performed the following will be false:, Object Craves other object,
↪  Province object, Harmony.

[STATEMENT]
As initial conditions I have that, object b craves object c, harmony, planet object a,
↪  planet object c, planet object d, province object a, province object b and province
↪  object d.
My goal is to have that object c craves object b.

What is the plan to achieve my goal? Just give the actions in the plan.
```

## A.6 One shot Mystery Blocksworld with TMK Prompt Example

### One shot Classic Blocksworld with TMK Prompt Example

```
You must adhere strictly to the JSON below, paying attention to the rules, ensuring to
↪  use only moves spelt out in the JSON to achieve the final plan.
{
    "Goals": [
        {
            "name": "AttackObject",
            "description": "Attack an object from the planet.",
            "inputParameters": [
                "object",
                "configuration"
            ],
            "outputParameters": [
                "newConfiguration"
            ],
            "given": {
                "Planet(object)": true,
                "Province(object)": true,
                "Harmony": true
            },
            "makes": {
                "Pain(object)": true,
                "Province(object)": false,
                "Planet(object)": false,
                "Harmony": false
            },
            "mechanism": "AttackObjectMechanism"
        },
        {
            "name": "SuccumbObject",
            "description": "Succumb a Pain object onto the planet.",
            "inputParameters": [
                "object",
                "configuration"
            ],
            "outputParameters": [
                "newConfiguration"
            ],
            "given": {
                "Pain(object)": true
            },
            "makes": {
                "Planet(object)": true,
                "Province(object)": true,
                "Harmony": true,
                "Pain(object)": false
            },
            "mechanism": "SuccumbObjectMechanism"
        },
        {
            "name": "OvercomeObject",
            "description": "Overcome a Pain object onto another Province object.",
            "inputParameters": [
                "objectToOvercome",
                "objectTarget",
                "configuration"
            ],
            "outputParameters": [
                "newConfiguration"
```

```
1080
1081                 ],
                     "given": {
1082                     "Pain(objectToOvercome)": true,
                         "Province(objectTarget)": true
1083                 },
                     "makes": {
1084                     "ObjectCraves(objectToOvercome, objectTarget)": true,
1085                     "Province(objectToOvercome)": true,
                         "Province(objectTarget)": false,
1086                     "Harmony": true,
1087                     "Pain(objectToOvercome)": false
1088                 },
                     "mechanism": "OvercomeObjectMechanism"
1089             },
                 {
1090                 "name": "FeastObject",
1091                 "description": "Feast an object from on top of another object (objectFrom).",
                     "inputParameters": [
1092                     "objectToFeast",
1093                     "objectFrom",
                         "configuration"
1094                 ],
1095                 "outputParameters": [
                         "newConfiguration"
1096                 ],
1097                 "given": {
                         "ObjectCraves(objectToFeast, objectFrom)": true,
1098                     "Province(objectToFeast)": true,
1099                     "Harmony": true
                     },
1100                 "makes": {
1101                     "Pain(objectToFeast)": true,
                         "Province(objectFrom)": true,
1102                     "ObjectCraves(objectToFeast, objectFrom)": false,
                         "Province(objectToFeast)": false,
1103                     "Harmony": false
1104                 },
                     "mechanism": "FeastObjectMechanism"
1105             }
            ],
1106         "Mechanisms": [
1107             {
                     "name": "AttackObjectMechanism",
1108                 "description": "Attack {object}.",
1109                 "inputParameters": [
1110                     "object",
                         "configuration"
1111                 ],
                     "outputParameters": [
1112                     "newConfiguration"
1113                 ],
                     "type": "operation",
1114                 "requires": {
1115                     "Planet(object)": true,
1116                     "Province(object)": true,
                         "Harmony": true
1117                 },
                     "provides": {
1118                     "Pain(object)": true,
1119                     "Harmony": false,
                         "Planet(object)": false,
1120                     "Province(object)": false
                     },
1121                 "process": "Remove Planet(object), add Pain(object), remove Province(object),
1122             ↪   set NOT Harmony"
1123             },
                 {
1124                 "name": "SuccumbObjectMechanism",
                     "description": "Succumb {object}.",
1125                 "inputParameters": [
1126                     "object",
                         "configuration"
1127                 ],
                     "outputParameters": [
1128                     "newConfiguration"
1129                 ],
                     "type": "operation",
1130                 "requires": {
1131                     "Pain(object)": true
                     },
1132                 "provides": {
1133                     "Planet(object)": true,
```

```
1134              "Harmony": true,
1135              "Province(object)": true,
1136              "Pain(object)": false
            },
1137          "process": "Remove Pain(object), add Planet(object), add Province(object),
1138          ↪   set Harmony"
        },
1139        {
1140              "name": "OvercomeObjectMechanism",
1141              "description": "Overcome {objectToOvercome} on {objectTarget}.",
              "inputParameters": [
1142              "objectToOvercome",
                "objectTarget",
1143              "configuration"
            ],
1144          "outputParameters": [
1145              "newConfiguration"
            ],
1146          "type": "operation",
1147          "requires": {
1148              "Pain(objectToOvercome)": true,
                "Province(objectTarget)": true
1149          },
            "provides": {
1150              "ObjectCraves(objectToOvercome, objectTarget)": true,
1151              "Harmony": true,
1152              "Province(objectToOvercome)": true,
                "Province(objectTarget)": false,
1153              "Pain(objectToOvercome)": false
            },
1154          "process": "Remove Pain(objectToOvercome), add ObjectCraves(objectToOvercome,
1155          ↪   objectTarget), add Province(objectToOvercome), remove
1156          ↪   Province(objectTarget), set Harmony"
        },
1157        {
1158              "name": "FeastObjectMechanism",
              "description": "Feast {objectToFeast} from {objectFrom}.",
1159          "inputParameters": [
1160              "objectToFeast",
                "objectFrom",
1161              "configuration"
            ],
1162          "outputParameters": [
1163              "newConfiguration"
            ],
1164          "type": "operation",
1165          "requires": {
                "ObjectCraves(objectToFeast, objectFrom)": true,
1166              "Province(objectToFeast)": true,
                "Harmony": true
1167          },
1168          "provides": {
                "Pain(objectToFeast)": true,
1169              "Province(objectFrom)": true,
                "ObjectCraves(objectToFeast, objectFrom)": false,
1170              "Province(objectToFeast)": false,
1171              "Harmony": false
            },
1172          "process": "Remove ObjectCraves(objectToFeast, objectFrom), add
1173          ↪   Pain(objectToFeast), add Province(objectFrom), remove
1174          ↪   Province(objectToFeast), set NOT Harmony"
        }
1175    ],
    "Knowledge": {
1176      "Concepts": [
1177        {
              "name": "object",
1178          "description": "An object in this domain that when is Province can be
1179          ↪   Attack from the Planet, Succumb onto the Planet, Overcome onto
              ↪   another object, or Feast from another object."
1180        },
1181        {
              "name": "hand",
1182          "description": "The manipulator that can Attack a object on the Planet,
1183          ↪   Succumb an object onto the Planet, Overcome onto another object, or
              ↪   Feast a Province object from another object. When hand is Harmony it
1184          ↪   can Attack, or Feast an object. When hand is Pain object, the same
              ↪   object can Succumb an object onto the Planet or Overcome another
1185          ↪   Province object."
1186        },
1187        {
```

```
                    "name": "configuration",
                    "description": "Complete state of this domain world."
                }
            ],
            "Relations": [
                {
                    "name": "ObjectCraves",
                    "description": "Binary Predicate: Relates an object to what it is on top
                    ↪   of (another object), represented as ObjectCraves(object,
                    ↪   anotherObject)."
                },
                {
                    "name": "Pain",
                    "description": "Unary Predicate: Relates the hand to the Pain object by
                    ↪   setting Pain(object)."
                },
                {
                    "name": "Planet",
                    "description": "Unary Predicate: The surface where objects can be Attack
                    ↪   from or Succumb onto, represented as Planet(object)."
                },
                {
                    "name": "Province",
                    "description": "Unary Predicate: An object is Province if no other object
                    ↪   is on top of it, represented as Province(object)."
                },
                {
                    "name": "Harmony",
                    "description": "Predicate: The hand is Harmony, not Pain any object,
                    ↪   represented as Harmony."
                }
            ]
        }
}
Below, within [Plan] and [Plan End], is the format you will use for the answer. The first
↪   one is an example. Focus on only the second plan.

[STATEMENT]
As initial conditions I have that, object b craves object c, harmony, planet object a,
↪   planet object c, planet object d, province object a, province object b and province
↪   object d.
My goal is to have that object c craves object b.

My plan is as follows:

[PLAN]
feast object b from object c
succumb object b
attack object c
overcome object c from object b
[PLAN END]

[STATEMENT]
As initial conditions I have that: object a craves object b, object d craves object c,
↪   harmony, planet object b, planet object c, province object a and province object d.
My goal is for the following to be true: object c craves object a.

My plan is as follows:

[PLAN]
```

## A.7  ONE SHOT RANDOM BLOCKSWORLD WITH PLAIN TEXT EXAMPLE

**One shot Random Blocksworld with Plain Text**

```
I am playing with a set of objects. Here are the actions I can do

1jpkithdyjmlikck object
xptxjrdkbi3pqsqr object from another object
9big8ruzarkkquyu object
2ijg9q8swj2shjel object from another object

I have the following restrictions on my actions:
To perform 1jpkithdyjmlikck action, the following facts need to be true: aqcjuuehivl8auwt
↪   object, 51nbwlachmfartjn object, 3covmuy4yrjthijd.
```

```
Once 1jpkithdyjmlikck action is performed the following facts will be true:
↪   gk5asm3f7u1fekpj object.
Once 1jpkithdyjmlikck action is performed the following facts will be false:
↪   aqcjuuehivl8auwt object, 51nbwlachmfartjn object, 3covmuy4yrjthijd.
To perform 9big8ruzarkkquyu action, the following facts need to be true: gk5asm3f7u1fekpj
↪   object.
Once 9big8ruzarkkquyu action is performed the following facts will be true:
↪   aqcjuuehivl8auwt object, 51nbwlachmfartjn object, 3covmuy4yrjthijd.
Once 9big8ruzarkkquyu action is performed the following facts will be false:
↪   gk5asm3f7u1fekpj object.
To perform 2ijg9q8swj2shjel action, the following needs to be true: aqcjuuehivl8auwt
↪   other object, gk5asm3f7u1fekpj object.
Once 2ijg9q8swj2shjel action is performed the following will be true: 3covmuy4yrjthijd,
↪   aqcjuuehivl8auwt object, Object 4DMF1cMTYXGSP94G other object.
Once 2ijg9q8swj2shjel action is performed the following will be false: aqcjuuehivl8auwt
↪   other object, gk5asm3f7u1fekpj object.
To perform xptxjrdkbi3pqsqr action, the following needs to be true: Object
↪   4DMF1cMTYXGSP94G other object, aqcjuuehivl8auwt object, 3covmuy4yrjthijd.
Once xptxjrdkbi3pqsqr action is performed the following will be true: gk5asm3f7u1fekpj
↪   object, aqcjuuehivl8auwt other object.
Once xptxjrdkbi3pqsqr action is performed the following will be false:, Object
↪   4DMF1cMTYXGSP94G other object, aqcjuuehivl8auwt object, 3covmuy4yrjthijd.

[STATEMENT]
As initial conditions I have that: aqcjuuehivl8auwt object a, aqcjuuehivl8auwt object b,
↪   aqcjuuehivl8auwt object d, 3covmuy4yrjthijd, object b 4dmf1cmtyxgsp94g object c,
↪   51nbwlachmfartjn object a, 51nbwlachmfartjn object c and 51nbwlachmfartjn object d.
My goal is for the following to be true: object c 4dmf1cmtyxgsp94g object b.

My plan is as follows:

[PLAN]
xptxjrdkbi3pqsqr object b from object c
9big8ruzarkkquyu object b
1jpkithdyjmlikck object c
2ijg9q8swj2shjel object c from object b
[PLAN END]

[STATEMENT]
As initial conditions I have that: aqcjuuehivl8auwt object a, aqcjuuehivl8auwt object d,
↪   3covmuy4yrjthijd, object a 4dmf1cmtyxgsp94g object b, object d 4dmf1cmtyxgsp94g
↪   object c, 51nbwlachmfartjn object b and 51nbwlachmfartjn object c.
My goal is for the following to be true: object c 4dmf1cmtyxgsp94g object a.

My plan is as follows:

[PLAN]
```

## A.8   ZERO SHOT RANDOM BLOCKSWORLD WITH PLAIN TEXT EXAMPLE

### Zero shot Random Blocksworld with Plain Text

```
I am playing with a set of objects. Here are the actions I can do

1jpkithdyjmlikck object
xptxjrdkbi3pqsqr object from another object
9big8ruzarkkquyu object
2ijg9q8swj2shjel object from another object

I have the following restrictions on my actions:
To perform 1jpkithdyjmlikck action, the following facts need to be true: aqcjuuehivl8auwt
↪   object, 51nbwlachmfartjn object, 3covmuy4yrjthijd.
Once 1jpkithdyjmlikck action is performed the following facts will be true:
↪   gk5asm3f7u1fekpj object.
Once 1jpkithdyjmlikck action is performed the following facts will be false:
↪   aqcjuuehivl8auwt object, 51nbwlachmfartjn object, 3covmuy4yrjthijd.
To perform 9big8ruzarkkquyu action, the following facts need to be true: gk5asm3f7u1fekpj
↪   object.
Once 9big8ruzarkkquyu action is performed the following facts will be true:
↪   aqcjuuehivl8auwt object, 51nbwlachmfartjn object, 3covmuy4yrjthijd.
Once 9big8ruzarkkquyu action is performed the following facts will be false:
↪   gk5asm3f7u1fekpj object.
To perform 2ijg9q8swj2shjel action, the following needs to be true: aqcjuuehivl8auwt
↪   other object, gk5asm3f7u1fekpj object.
Once 2ijg9q8swj2shjel action is performed the following will be true: 3covmuy4yrjthijd,
↪   aqcjuuehivl8auwt object, Object 4DMF1cMTYXGSP94G other object.
Once 2ijg9q8swj2shjel action is performed the following will be false: aqcjuuehivl8auwt
↪   other object, gk5asm3f7u1fekpj object.
```

```
To perform xptxjrdkbi3pqsqr action, the following needs to be true: Object
↪  4DMF1cMTYXGSP94G other object, aqcjuuehivl8auwt object, 3covmuy4yrjthijd.
Once xptxjrdkbi3pqsqr action is performed the following will be true: gk5asm3f7u1fekpj
↪  object, aqcjuuehivl8auwt other object.
Once xptxjrdkbi3pqsqr action is performed the following will be false:, Object
↪  4DMF1cMTYXGSP94G other object, aqcjuuehivl8auwt object, 3covmuy4yrjthijd.

[STATEMENT]
As initial conditions I have that: aqcjuuehivl8auwt object a, aqcjuuehivl8auwt object b,
↪  aqcjuuehivl8auwt object d, 3covmuy4yrjthijd, object b 4dmf1cmtyxgsp94g object c,
↪  51nbwlachmfartjn object a, 51nbwlachmfartjn object c and 51nbwlachmfartjn object d.
My goal is for the following to be true: object c 4dmf1cmtyxgsp94g object b.

To solve the problem, you will have to provide which actions to take from the initial
↪  conditions and in which order in order to achieve the goal conditions. Provide the
↪  plan by giving the action names along with the objects \"ACTION_NAME OBJECTS\".
↪  Provide the plan between these two tags [PLAN] and [PLAN END].
```

## A.9 ONE SHOT RANDOM BLOCKSWORLD WITH TMK PROMPT EXAMPLE

**One shot Random Blocksworld with TMK Prompt**

```
You must adhere strictly to the JSON below, paying attention to the rules, ensuring to
↪  use only moves spelt out in the JSON to achieve the final plan.
{
    "Goals": [

        {
            "name": "1jpkithdyjmlikck",
            "description": "1jpkithdyjmlikck an object from the 51nbwlachmfartjn.",
            "inputParameters": [
                "object",
                "configuration"
            ],
            "outputParameters": [
                "newConfiguration"
            ],
            "given": {
                "51nbwlachmfartjn(object)": true,
                "aqcjuuehivl8auwt(object)": true,
                "3covmuy4yrjthijd": true
            },
            "makes": {
                "gk5asm3f7u1fekpj(object)": true,
                "aqcjuuehivl8auwt(object)": false,
                "51nbwlachmfartjn(object)": false,
                "3covmuy4yrjthijd": false
            },
            "mechanism": "1jpkithdyjmlikckMechanism"
        },
        {
            "name": "9big8ruzarkkquyu",
            "description": "9big8ruzarkkquyu a gk5asm3f7u1fekpj object onto the
            ↪  51nbwlachmfartjn.",
            "inputParameters": [
                "object",
                "configuration"
            ],
            "outputParameters": [
                "newConfiguration"
            ],
            "given": {
                "gk5asm3f7u1fekpj(object)": true
            },
            "makes": {
                "51nbwlachmfartjn(object)": true,
                "aqcjuuehivl8auwt(object)": true,
                "3covmuy4yrjthijd": true,
                "gk5asm3f7u1fekpj(object)": false
            },
            "mechanism": "9big8ruzarkkquyuMechanism"
        },
        {
            "name": "2ijg9q8swj2shjel",
            "description": "2ijg9q8swj2shjel a gk5asm3f7u1fekpj object (objectToOvercome)
            ↪  onto another aqcjuuehivl8auwt object (objectTarget).",
            "inputParameters": [
                "objectToOvercome",
```

```
                    "objectTarget",
                    "configuration"
                ],
                "outputParameters": [
                    "newConfiguration"
                ],
                "given": {
                    "gk5asm3f7u1fekpj(objectToOvercome)": true,
                    "aqcjuuehivl8auwt(objectTarget)": true
                },
                "makes": {
                    "4dmf1cmtyxgsp94g(objectToOvercome, objectTarget)": true,
                    "aqcjuuehivl8auwt(objectToOvercome)": true,
                    "aqcjuuehivl8auwt(objectTarget)": false,
                    "3covmuy4yrjthijd": true,
                    "gk5asm3f7u1fekpj(objectToOvercome)": false
                },
                "mechanism": "2ijg9q8swj2shjelMechanism"
            },
            {
                "name": "xptxjrdkbi3pqsqr",
                "description": "xptxjrdkbi3pqsqr an object from on top of another object
                ↪ (objectFrom).",
                "inputParameters": [
                    "objectToFeast",
                    "objectFrom",
                    "configuration"
                ],
                "outputParameters": [
                    "newConfiguration"
                ],
                "given": {
                    "4dmf1cmtyxgsp94g(objectToFeast, objectFrom)": true,
                    "aqcjuuehivl8auwt(objectToFeast)": true,
                    "3covmuy4yrjthijd": true
                },
                "makes": {
                    "gk5asm3f7u1fekpj(objectToFeast)": true,
                    "aqcjuuehivl8auwt(objectFrom)": true,
                    "4dmf1cmtyxgsp94g(objectToFeast, objectFrom)": false,
                    "aqcjuuehivl8auwt(objectToFeast)": false,
                    "3covmuy4yrjthijd": false
                },
                "mechanism": "xptxjrdkbi3pqsqrMechanism"
            }
        ],
        "Mechanisms": [
            {
                "name": "1jpkithdyjmlikckMechanism",
                "description": "1jpkithdyjmlikck {object}.",
                "inputParameters": [
                    "object",
                    "configuration"
                ],
                "outputParameters": [
                    "newConfiguration"
                ],
                "type": "operation",
                "requires": {
                    "51nbwlachmfartjn(object)": true,
                    "aqcjuuehivl8auwt(object)": true,
                    "3covmuy4yrjthijd": true
                },
                "provides": {
                    "gk5asm3f7u1fekpj(object)": true,
                    "3covmuy4yrjthijd": false,
                    "51nbwlachmfartjn(object)": false,
                    "aqcjuuehivl8auwt(object)": false
                },
                "process": "Remove 51nbwlachmfartjn(object), add gk5asm3f7u1fekpj(object),
                ↪  remove aqcjuuehivl8auwt(object), set NOT 3covmuy4yrjthijd"
            },
            {
                "name": "9big8ruzarkkquyuMechanism",
                "description": "9big8ruzarkkquyu {object}.",
                "inputParameters": [
                    "object",
                    "configuration"
                ],
                "outputParameters": [
```

```
1404                    "newConfiguration"
1405                ],
                    "type": "operation",
1406                "requires": {
1407                    "gk5asm3f7u1fekpj(object)": true
1408                },
                    "provides": {
1409                    "51nbwlachmfartjn(object)": true,
1410                    "3covmuy4yrjthijd": true,
                    "aqcjuuehivl8auwt(object)": true,
1411                    "gk5asm3f7u1fekpj(object)": false
1412                },
                    "process": "Remove gk5asm3f7u1fekpj(object), add 51nbwlachmfartjn(object),
1413            ↪    add aqcjuuehivl8auwt(object), set 3covmuy4yrjthijd"
1414            },
                {
1415                "name": "2ijg9q8swj2shjelMechanism",
1416                "description": "2ijg9q8swj2shjel {objectToOvercome} on {objectTarget}.",
                    "inputParameters": [
1417                    "objectToOvercome",
1418                    "objectTarget",
                    "configuration"
1419                ],
                    "outputParameters": [
1420                    "newConfiguration"
1421                ],
                    "type": "operation",
1422                "requires": {
1423                    "gk5asm3f7u1fekpj(objectToOvercome)": true,
                    "aqcjuuehivl8auwt(objectTarget)": true
1424                },
                    "provides": {
1425                    "4dmf1cmtyxgsp94g(objectToOvercome, objectTarget)": true,
1426                    "3covmuy4yrjthijd": true,
                    "aqcjuuehivl8auwt(objectToOvercome)": true,
1427                    "aqcjuuehivl8auwt(objectTarget)": false,
1428                    "gk5asm3f7u1fekpj(objectToOvercome)": false
                },
1429                "process": "Remove gk5asm3f7u1fekpj(objectToOvercome), add
1430            ↪    4dmf1cmtyxgsp94g(objectToOvercome, objectTarget), add
                ↪    aqcjuuehivl8auwt(objectToOvercome), remove
1431            ↪    aqcjuuehivl8auwt(objectTarget), set 3covmuy4yrjthijd"
1432            },
                {
1433                "name": "xptxjrdkbi3pqsqrMechanism",
1434                "description": "xptxjrdkbi3pqsqr {objectToFeast} from {objectFrom}.",
                    "inputParameters": [
1435                    "objectToFeast",
1436                    "objectFrom",
                    "configuration"
1437                ],
                    "outputParameters": [
1438                    "newConfiguration"
1439                ],
                    "type": "operation",
1440                "requires": {
1441                    "4dmf1cmtyxgsp94g(objectToFeast, objectFrom)": true,
                    "aqcjuuehivl8auwt(objectToFeast)": true,
1442                    "3covmuy4yrjthijd": true
1443                },
                    "provides": {
1444                    "gk5asm3f7u1fekpj(objectToFeast)": true,
                    "aqcjuuehivl8auwt(objectFrom)": true,
1445                    "4dmf1cmtyxgsp94g(objectToFeast, objectFrom)": false,
1446                    "aqcjuuehivl8auwt(objectToFeast)": false,
                    "3covmuy4yrjthijd": false
1447                },
                    "process": "Remove 4dmf1cmtyxgsp94g(objectToFeast, objectFrom), add
1448            ↪    gk5asm3f7u1fekpj(objectToFeast), add aqcjuuehivl8auwt(objectFrom), remove
                ↪    aqcjuuehivl8auwt(objectToFeast), set NOT 3covmuy4yrjthijd"
1449            }
        ],
1450        "Knowledge": {
1451            "Concepts": [
1452                {
                    "name": "object",
1453                "description": "An object in this domain that when is aqcjuuehivl8auwt
1454            ↪    can be 1jpkithdyjmlikck from the 51nbwlachmfartjn, 9big8ruzarkkquyu
                ↪    onto the 51nbwlachmfartjn, 2ijg9q8swj2shjel onto another object, or
1455            ↪    xptxjrdkbi3pqsqr from another object."
1456            },
1457
```

```
            {
                "name": "hand",
                "description": "The manipulator that can 1jpkithdyjmlikck an object from
                ↪  the 51nbwlachmfartjn, 9big8ruzarkkquyu an object onto the
                ↪  51nbwlachmfartjn, 2ijg9q8swj2shjel onto another object, or
                ↪  xptxjrdkbi3pqsqr an aqcjuuehivl8auwt object from another object. When
                ↪  hand is 3covmuy4yrjthijd it can 1jpkithdyjmlikck or xptxjrdkbi3pqsqr
                ↪  an object. When hand is gk5asm3f7u1fekpj object, the same object can
                ↪  9big8ruzarkkquyu an object onto the 51nbwlachmfartjn or
                ↪  2ijg9q8swj2shjel another aqcjuuehivl8auwt object."
            },
            {
                "name": "configuration",
                "description": "Complete state of this domain world."
            },
            {
                "name": "isWellFormed",
                "description": "Configuration follows all domain rules."
            },
            {
                "name": "matches",
                "description": "Two configurations are identical."
            }
        ],
        "Relations": [
            {
                "name": "4dmf1cmtyxgsp94g",
                "description": "Binary Predicate: Relates an object to what it is on top
                ↪  of (another object), represented as 4dmf1cmtyxgsp94g(object,
                ↪  anotherObject)."
            },
            {
                "name": "gk5asm3f7u1fekpj",
                "description": "Unary Predicate: Relates the hand to the held object by
                ↪  setting gk5asm3f7u1fekpj(object)."
            },
            {
                "name": "51nbwlachmfartjn",
                "description": "Unary Predicate: The surface where objects can be picked
                ↪  up from or put down onto, represented as 51nbwlachmfartjn(object)."
            },
            {
                "name": "aqcjuuehivl8auwt",
                "description": "Unary Predicate: An object is clear if no other object is
                ↪  on top of it, represented as aqcjuuehivl8auwt(object)."
            },
            {
                "name": "3covmuy4yrjthijd",
                "description": "Zero-arity Predicate: The hand is empty, represented as
                ↪  3covmuy4yrjthijd."
            }
        ]
    }
}
Below, within [Plan] and [Plan End], is the format you will use for the answer. The first
↪  one is an example. Focus on only the second plan.

[STATEMENT]
As initial conditions I have that: aqcjuuehivl8auwt object a, aqcjuuehivl8auwt object b,
↪  aqcjuuehivl8auwt object d, 3covmuy4yrjthijd, object b 4dmf1cmtyxgsp94g object c,
↪  51nbwlachmfartjn object a, 51nbwlachmfartjn object c and 51nbwlachmfartjn object d.
My goal is for the following to be true: object c 4dmf1cmtyxgsp94g object b.

My plan is as follows:

[PLAN]
xptxjrdkbi3pqsqr object b from object c
9big8ruzarkkquyu object b
1jpkithdyjmlikck object c
2ijg9q8swj2shjel object c from object b
[PLAN END]

[STATEMENT]
As initial conditions I have that: aqcjuuehivl8auwt object a, aqcjuuehivl8auwt object d,
↪  3covmuy4yrjthijd, object a 4dmf1cmtyxgsp94g object b, object d 4dmf1cmtyxgsp94g
↪  object c, 51nbwlachmfartjn object b and 51nbwlachmfartjn object c.
My goal is for the following to be true: object c 4dmf1cmtyxgsp94g object a.

My plan is as follows:

[PLAN]
```

## A.10 DECLARATION

Declaration of LLM usage: LLMs were used to discover and trace planbench code issues, formatting of figures and tables, retrieval and discovery of existing research papers (mixed with google search). In the rebuttal phase, language models were used suggest edits of existing content.

