# OpenReview forum: "Knowledge Model Prompting Increases LLM Performance on Planning Tasks"
_ICLR.cc/2026/Conference — Submitted to ICLR 2026_

### Official Review · Reviewer_MjAL · 2025-10-29

**Soundness:** 2
**Presentation:** 1
**Contribution:** 2
**Rating:** 2
**Confidence:** 3

**Summary:**

This paper introduces a prompting approach based on the Task–Method–Knowledge (TMK) framework, originally from educational research. The authors argue that LLMs often fail on multi-step planning tasks because typical prompts provide only shallow textual goals. TMK decomposes each problem into structured layers: task (goal and conditions), method (procedure), and knowledge (objects and relations). The authors uses PlanBench’s Blocksworld domain in a TMK-formatted json and compare it with the plain prompts on GPT family (GPT-4, GPT-4o, o1-mini, o1, and GPT-5). Results show modest but consistent improvements.

**Strengths:**

• The paper introduces a simple and interpretable idea: representing planning tasks in a structured TMK format may align with how procedural knowledge is expressed in model pre-training data

• The approach is prompt-based and requires no fine-tuning or external resources, making it easy to reproduce and extend.

• The empirical trend (larger gains for weaker models) is intuitive and suggests the TMK structure provides helpful inductive bias.

**Weaknesses:**

• The experiments are narrow: only one domain and one benchmark family. Claims about general planning improvement are therefore not well supported.

• There are no comparisons with so many other structured prompting methods (eg CoS, ReAct, least-to-most, chain-of-thought scaffolding and so on). It is unclear whether TMK offers advantages beyond simply using more structured json templates.

• The explanation of why TMK helps remains speculative. No ablation isolates whether improvements come from the educational knowledge model framing or just from better formatting and key-word cues.

**Questions:**

The writing quality and technical presentation are weak, with many typos, missing citations, and incorrect or inconsistent latex usage (eg mismatched quotes and unescaped underscores). The paper would need careful proofreading and formatting cleanup.

---

> ### Author Response · Authors · 2025-11-21
> **Response to Reviewer MjAL**
>
> Thank you for your fair and constructive review. We agree with your assessment of the strengths and weaknesses of our paper and have addressed your points in our revision.
>
> **Narrow experiments:** We agree that the current experiments are narrow. We have added a "Limitations" section (5.3) where we acknowledge this and discuss the need to evaluate our method on more domains and benchmarks to support our claims about general planning improvement.
>
> **No comparisons with other structured prompting methods:** This is a valid point. You offered two main methods for expanding our research to become comprehensive enough to publish. One suggested we investigate other models, the other suggested we investigate other method (chain of thought, React). Obviously doing both of these could become prohibitively expensive.
>
> * The literature suggests that chain of thought does not improve planning tasks for our domain, as outlined in the paper. Further additional research into CoT and ReAct document similar challenges to the ones outlined in our paper [5][6]. We reworked the literature discussion to reinforce this point in section 2.1.
>
> * May we inquire how many models would be adequate coverage, and if open-source models would be acceptable?
>
> **Why TMK helps remains speculative:** We have expanded our discussion in Section 5.2 to provide a more detailed and grounded explanation of why we believe the TMK framework helps. We have also updated our hypothesis and included an important new finding [7] related to the structural similarity of TMK to code that together with scores validates the hypothesis.
>
> **Writing and presentation:** We have proofread and edited the manuscript to address the issues you raised regarding typos, missing citations, and inconsistent LaTeX usage and will continue to do so throughout.

---

### Official Review · Reviewer_NaSB · 2025-10-31

**Soundness:** 1
**Presentation:** 1
**Contribution:** 1
**Rating:** 0
**Confidence:** 5

**Summary:**

The paper studies integrating the Task-Knowledge-Method (TMK) framework into prompting for LLM reasoning tasks. It focused on the blocksworld task in Planbench and experimented with OpenAI models.

**Strengths:**

The paper shows the potential of applying a prompting technique to improve the performance of models on the blocksworld task.

**Weaknesses:**

1. The reviewer finds it difficult to understand what is the core idea/contribution of TMK prompting method proposed in the paper. The paper did poorly in explaining the proposed prompting method.
2. The paper proposes a prompting method, yet there are no examples of the prompt in the paper.
3. Experimental evidence of the advantage of the proposed method is very limited: only OpenAI models, only one task (blocksworld), and many numbers are missing (Table 2)

**Questions:**

The presentation of the paper needs a lot more work, to list a few:
1. Missing citation in line 39.
2. Missing space in line 50.
3. Missing citation in line 394.
4. A formatting suggestion: most citations in the paper should use `\citep{}` instead of `\cite{}`.
5. Figure 1 is too big, and the resolution of the image is too low.

---

> ### Author Response · Authors · 2025-11-21
> **Response to Reviewer NaSB**
>
> We appreciate your detailed feedback on our paper. We have made revisions to improve the presentation and clarity of our work. Below addresses the weakness points sequentially.
>
> **Difficulty understanding the core idea of TMK:** We have revised the introduction and Section 2.1 to provide a clearer and more comprehensive explanation of the Task-Method-Knowledge (TMK) framework and its application to LLM prompting. In addition to the source files, we have also added a full prompt example in the Appendix to make the method more concrete.
>
> **No examples of the prompt in the paper:** Per above, we have now included TMK prompt in the Appendix.
>
> **Limited experimental evidence:** We acknowledge the limitations of our current experiments. We have updated Table 2 to be more complete and have provided a clearer explanation of our results. We have also added to the "Limitations" section (5.3) where we discuss the need for experiments with more models and tasks and outline this as a direction for future work.
>
> **Presentation and formatting issues:** We have addressed all the formatting issues you pointed out, including the missing citations, missing space, and the use of `\citep`. We have also improved the resolution of Figure 1 and adjusted its size, expanded it further to show the hierarchical nature of TMK.

---

### Official Review · Reviewer_CH5Z · 2025-11-01

**Soundness:** 1
**Presentation:** 1
**Contribution:** 1
**Rating:** 0
**Confidence:** 3

**Summary:**

The paper proposes TMK, a framework to capture specific reasoning structures in LLM reasoning tasks. It features explicit task decomposition, and benefits the LLM reasoning tasks. It proposes a PlanBench benchmark to conduct experiments.

**Strengths:**

The method focuses on the key problem of long reasoning LLMs, which does not have clear task decomposition during thinking.

**Weaknesses:**

+ Many fields are missing in Table 2, especially Plain Text + One Shot. Note that it's unfair to compare the other two columns (TMK + One Shot vs Plain Text + Zero Shot). The paper does not have other main results.
+ It does not make sense to replace standard description of blockworlds into irrelevant mystery or random words. No LLM learns it during pre-training, nor people will use those words to describe tasks, nor they will use LLMs like this.
+ It's not easy to understand what TMK is doing (lacking a concrete example of the prompt/formatting).
+ The method is only experimented on blockworlds, which is a toy dataset easily solvable by non-machine learning algorithms like DFS. It's unknown whether the method can be generalized to other meaningful domains, such as mathematical, logical, legal, scientific reasonings.

**Questions:**

+ There is only 1 item in the list (Line 054)
+ Gpt5 -> GPT-5 (Line 308)
+ Unknown citation (?) (Line 394)

---

> ### Author Response · Authors · 2025-11-21
> **Response to Reviewer CH5Z**
>
> We thank you for your review and the points you raised. We have addressed your concerns in the revised manuscript. Below addresses the weakness points sequentially.
>
> **Missing fields in Table 2:** Thank you for highlighting the gaps in Table 2. We have conducted the additional experiments for o1 on the Random dataset using plain text prompts. As shown in the updated Table 2, the o1 model achieved only 31.5% with plain text but surged to 97.33% with TMK. This 65.8% gain provides even stronger empirical support for our hypothesis that TMK enables symbolic manipulation in reasoning models, we have updated our hypothsis to reflect that. We have also clarified that 'NA' values for o1-preview are due to the model's deprecation by OpenAI.
>
> **Zero vs Oneshot:** Thank you for raising this concern. While the original PlanBench paper utilizes one-shot prompts, the public leaderboard relies on zero-shot. We chose to compare against the higher-performing baseline (typically zero-shot) to ensure rigor. Additionally, using a one-shot approach for TMK allowed us to align outputs exactly with PlanBench's [1] requirements while minimizing formatting instructions.
>
> Literature suggests that zero-shot often outperforms few-shot prompting for LLMs [2]. To ensure a fair comparison, we conducted our own sampling of One-Shot vs. Zero-Shot performance for plain text on select models. Our internal results confirmed findings in [3] that Zero-Shot prompts generally outperform One-Shot prompts in the Blocksworld domain. Consequently, Table 2 compares our TMK One-Shot results against the strongest plain-text configuration (whether Zero or One-Shot) to provide the most rigorous baseline possible. While budget constraints prevented an exhaustive one-shot campaign for every model, our sampling aligns with prior PlanBench studies [3][4], which demonstrates no statistically significant advantage for one-shot over zero-shot. We have updated Section 3.2 to reflect this.
>
> **Irrelevant mystery or random words:** We recognise we need to explain this better in the original submission. The use of "mystery" and "random" words is an established part of the PlanBench benchmark [4], which is designed to test an LLM's planning ability independent of semantic cues in the language. We have added a more detailed explanation of the PlanBench benchmark in Section 2.2 to clarify this for readers.
>
> **Lacking a concrete example of the prompt/formatting:** We agree that a concrete example is essential for understanding our method. We have now included a TMK prompt examples in the Appendix to provide a clear example of how the TMK framework is implemented.
>
> **Experiments only on blocksworld:** We acknowledge that our experiments are currently limited to the blocksworld domain. However the idea of planbench [4] is to test the planning ability of LLMs, it is precisely because we are able to solve blockworld that we are able to verify how well (or badly) LLMs perform. We agree that testing on other domains is important for demonstrating the generalizability of our method, the publicly available benchmarks [1] have only focused on blocksworld and its mystery and randomized variants. We have added a discussion of this limitation in Section 5.3 and have outlined our plans for future work to expand our experiments to other planning tasks (i.e. Logistics).

---

### Author Response · Authors · 2025-11-21
**General Thanks and Response for ICLR Reviewers**

Thank you for your time and valuable feedback on our paper. We have made substantial revisions to the manuscript to address your concerns regarding clarity, experimental scope, and the theoretical mechanism of our method.

The main changes include:

* **Updated Hypothesis & Mechanism:** We have refined our core hypothesis to frame TMK not just as a context provider, but as a symbolic steering mechanism. We now provide empirical evidence (via an identified "performance inversion" in the o1 model) showing that TMK shifts reasoning from linguistic approximation to formal symbolic manipulation and integrated literature we found to support that.
* **Expanded Experimental Data:** We have updated Table 2 to include missing baselines (Plain Text) for a fairer comparison. Most notably, we added results for the o1 model on Random Blocksworld, revealing a 65.8% performance gain (31.5% -> 97.33%) with TMK prompting.
* **Concrete Examples:** We added full examples of the prompt variations to the Appendix to clarify the implementation.
* **Presentation:** We have corrected all identified typos, formatting issues (including citation styles), and improved the resolution of Figure 1 in addition to expanding it.

We believe these revisions address the concerns raised by the reviewers and strengthens the paper. We have provided detailed responses to each reviewer's comments below and are primarily seeking clarification and deeper reviews to further improve on the paper.







**References:**

[1] Karthik Valmeekam. Llms-planning: An extensible benchmark for evaluating large language models on planning. https://github.com/karthikv792/LLMs-Planning, 2023. Accessed: 2025-09-24.

[2] Takeshi Kojima, Shixiang Shane Gu, Machel Reid, Yutaka Matsuo, and Yusuke Iwasawa. Large language models are zero-shot reasoners. *Advances in neural information processing systems*, 35:22199–22213, 2022.

[3] Karthik Valmeekam, Matthew Marquez, Sarath Sreedharan, and Subbarao Kambhampati. On the planning abilities of large language models-a critical investigation. *Advances in Neural Information Processing Systems*, 36:75993–76005, 2023c.

[4] Karthik Valmeekam, Matthew Marquez, Alberto Olmo, Sarath Sreedharan, and Subbarao Kambhampati. Planbench: An extensible benchmark for evaluating large language models on planning and reasoning about change. *Advances in Neural Information Processing Systems*, 36:38975–38987, 2023a.

[5] Siddhant Bhambri, Mudit Verma, and Subbarao Kambhampati. Do think tags really help llms plan? A critical evaluation of react-style prompting. *Transactions on Machine Learning Research*, 2025.

[6] Kaya Stechly, Karthik Valmeekam, and Subbarao Kambhampati. Chain of thoughtlessness: An analysis of cot in planning. *arXiv preprint arXiv:2405.04776*, 2024.

[7] Yongchao Chen, Harsh Jhamtani, Srinagesh Sharma, Chuchu Fan, and Chi Wang. Steering large language models between code execution and textual reasoning. *arXiv preprint arXiv:2410.03524*, 2024.

---

### Meta-Review · Area_Chair_ZSLQ · 2026-01-07

**Summary:**

This paper proposes using the Task-Method-Knowledge (TMK) framework from educational science as a prompting strategy to improve LLM performance on planning tasks. The authors evaluate on PlanBench's Blocksworld domain using OpenAI models, reporting a notable result where o1's accuracy on Random Blocksworld jumps from 31.5% to 97.3% with TMK prompting.

All three reviewers who provided substantive reviews expressed serious concerns. The unanimous issues centered on: extremely narrow experimental scope limited to one domain and one model family, missing baselines and incomplete data in results tables creating unfair comparisons, absence of concrete prompt examples in the main paper, no comparison with established structured prompting methods like CoT, ReAct, or least-to-most prompting, speculative mechanistic explanations without ablation studies, and poor presentation quality including missing citations, typos, and formatting errors.

**Reviewer Concerns:**

The authors addressed several issues: added the missing o1 baseline (revealing the dramatic 65.8 point improvement), included prompt examples in the Appendix, fixed presentation errors, and refined the hypothesis to frame TMK as a "symbolic steering mechanism."
Core concerns remain unresolved. Experimental scope stays narrow despite budget explanations. The one-shot TMK vs zero-shot baseline comparison remains potentially unfair. No ablations isolate whether gains come from TMK's structure versus simply providing more organized JSON formatting. Citing literature that CoT doesn't help planning doesn't substitute for direct comparisons with alternative prompting methods.

**Reviewer Scores:**

Given unanimous negative reception and unresolved core concerns, the paper would likely not reach acceptance threshold.

---

### Decision · Program_Chairs · 2026-01-26

Reject